# Global prevalence of nosocomial infection: A systematic review and meta-analysis

**Samira Raoofi**[1,2], **Fatemeh Pashazadeh Kan**[3], **Sima Rafiei**[4], **Zahra Hosseinipalangi**[3],
**Zahra Noorani Mejareh**[5], **Saghar Khani**[5], **Bahare Abdollahi**[5], **Fatemeh Seyghalani Talab**[6],
**Mohaddeseh Sanaei**[3], **Farnaz Zarabi**[7], **Yasamin Dolati**[3], **Niloofar Ahmadi**[3], **Neda Raoofi**[8],
**Yasamin Sarhadi**[3], **Maryam Masoumi**[9], **Batool sadat Hosseini**[7], **Negin Vali**[10],
**Negin Gholamali**[3], **Saba Asadi**[11], **Saba Ahmadi**[3], **Behrooz Ahmadi**[12], **Zahra Beiramy
Chomalu**[3], **Elnaz Asadollahi**[2], **Mona Rajabi**[6], **Dorsa Gharagozloo**[13], **Zahra Nejatifar**[6],
**Rana Soheylirad**[6], **Shabnam Jalali**[5], **Farnaz Aghajani**[3], **Mobina Navidriahy**[3],
**Sama Deylami**[3], **Mahmoud Nasiri**[14], **Mahsa Zareei**[2], **Zahra Golmohammadi**[2],
**Hamideh Shabani**[2], **Fatemeh Torabi**[2], **Hosein Shabaninejad**[15], **Ali Nemati**[2],
**Mohammad Amerzadeh**[4], **Aidin Aryankhesal**[16], **Ahmad Ghashghaee**[17]*

1 School of Health Management and Information Sciences, Iran University of Medical Sciences, Tehran, Iran,
2 Student Research Committee, School of Health Management and Information Sciences, Iran University of
Medical Sciences, Tehran, Iran, 3 Student Research Committee, School of Nursing and Midwifery, Iran
University of Medical Sciences, Tehran, Iran, 4 Social Determinants of Health Research Center, Research
Institute for Prevention of Non-Communicable Diseases, Qazvin University of Medical Sciences, Qazvin, Iran,
5 Student Research Committee, School of Medicine, Iran University of Medical Sciences, Tehran, Iran,
6 Social Determinants of Health Research Center, Qazvin University of Medical Sciences, Qazvin, Iran,
7 Department of Anesthesia, School of Allied Medical Sciences, Iran University of Medical Sciences, Tehran,
Iran, 8 Cardiovascular Research Center Kermanshah, Kermanshah, Iran, 9 Clinical Research and
Development Center, Qom University of Medical Sciences, Qom, Iran, 10 Shahid AkbarAbadi Clinical
Research Development unit (SHACRDU), Iran University of Medical Sciences, Tehran, Iran, 11 Health
Management and Economics Research Center, Health Management Research Institute, Iran University of
Medical Sciences, Tehran, Iran, 12 Clinical Research Development Center, Imam Ali Hospital, Kermanshah
University of Medical Sciences, Kermanshah, Iran, 13 Department of Molecular and Cellular Sciences,
Faculty of Advanced Sciences and Technology, Tehran Medical Sciences, Islamic Azad University, Tehran,
Iran, 14 Researcher at Toward Evidence (http://towardevidence.co.uk/), Glasgow, United Kingdom,
15 Population Health Sciences Institute (PHSI), Newcastle University, Newcastle, United Kingdom,
16 Department of Health Services Management, School of Health Management and Information Sciences,
Iran University of Medical Sciences, Tehran, Iran, 17 School of Medicine, Dentistry & Nursing, University of
Glasgow, Glasgow, United Kingdom

* ahmad.ghashghaee1996@gmail.com

HONG KONG

**Data Availability Statement:** All relevant data are
within the paper and its Supporting Information
files.

## Abstract

### Objectives

Hospital-acquired infections (HAIs) are significant problems as public health issues which
need attention. Such infections are significant problems for society and healthcare organiza-
tions. This study aimed to carry out a systematic review and a meta-analysis to analyze the
prevalence of HAIs globally.

### Methods

We conducted a comprehensive search of electronic databases including EMBASE, Sco-
pus, PubMed and Web of Science between 2000 and June 2021. We found 7031 articles.
After removing the duplicates, 5430 studies were screened based on the titles/ abstracts.

**Funding:** The authors received no specific funding for this work.

**Competing interests:** The authors have declared that no competing interests exist.

Then, we systematically evaluated the full texts of the 1909 remaining studies and selected 400 records with 29,159,630 participants for meta-analysis. Random-effects model was used for the analysis, and heterogeneity analysis and publication bias test were conducted.

## Results

The rate of universal HAIs was 0.14 percent. The rate of HAIs is increasing by 0.06 percent annually. The highest rate of HAIs was in the AFR, while the lowest prevalence were in AMR and WPR. Besides, AFR prevalence in central Africa is higher than in other parts of the world by 0.27 (95% CI, 0.22–0.34). Besides, E. coli infected patients more than other micro-organisms such as Coagulase-negative staphylococci, Staphylococcus spp. and Pseudomonas aeruginosa. In hospital wards, Transplant, and Neonatal wards and ICU had the highest rates. The prevalence of HAIs was higher in men than in women.

## Conclusion

We identified several essential details about the rate of HAIs in various parts of the world. The HAIs rate and the most common micro-organism were different in various contexts. However, several essential gaps were also identified. The study findings can help hospital managers and health policy makers identify the reason for HAIs and apply effective control programs to implement different plans to reduce the HAIs rate and the financial costs of such infections and save resources.

## Introduction

Hospital-acquired infections (HAIs) are significant problems which need serious attention worldwide. HAIs refer to a group of infections a patient does not have before admission to the hospital. HAIs do not even exist in the latency period; they occur upon arrival at the hospital or within 48–72 hours after admission to the hospital [1–4]. Nowadays, such infections are significant problems for societies and healthcare organizations. They prolong the treatment period and make both patients and health centers pay excessive costs, including increased drug intakes and tests [5]. Therefore, by preventing and reducing nosocomial infections, significant savings will be made in the costs imposed on health centers, the health system and society consequently [6].

Due to financial constraints, there are many problems in controlling HAIs in emerging countries. Besides the problems caused by the extension of hospital stay for the patient, HAIs can be transmitted to the patient's relatives through casual contacts and jeopardize their physical conditions [6]. Such infections are not limited to specific patients. They may occur to every patient or hospital employee and increase the mortality rate of hospitals [7].

According to studies, the most prevalent causes of HAIs include urinary tract infections (UTIs), respiratory tract infections (RTIs), circulatory system infections, and surgical site infections [8–10]. According to a report of the World Health Organization (WHO) on 55 hospitals in 14 countries, 8.7% of the hospitalized patients had HAIs, which were more prevalent in the Eastern Mediterranean Region and less prevalent in the West of the Pacific [11–13]. The prevalence rate of these infections was reported to be 5% in the North of America and some parts of Europe, and was about 40% in some Asian, Latin American, and African countries [14, 15]. According to the findings of a study conducted in Europe, the prevalence of HAIs

was nearly 2.9%. Medical interventions, poor health standards of the hospital environment, and poor personal hygiene of hospital staff and patients poor practice of personal hygiene among hospital staff and patients can cause HAIs [16]. However, the major/leading cause of HAIs is lack of compliance to health and safety guidelines of hospitals [17]. Although it is impossible to eliminate such infections even in the most advanced hospitals, standards and guidelines can be complied with the intention of reducing or managing them [18, 19]. Nowadays, with technological advances and high expectations of high quality care services, it is highly essential to analyze the frequency and causes of HAIs [20]. Therefore, it is necessary to know the prevalence rate of different HAIs to devise infection control programs in hospitals and help develop a reliable and effective plan. Lack of accurate data on the prevalence of HAIs makes the execution of control plans challenging and causes higher costs for health systems and patients [21, 22].

Due to the presence of developing and underdeveloped countries in the EMRO (the Eastern Mediterranean Regional Office of the World Health Organization), AFRO (African Regional Office of the World Health Organization) and other countries with high prevalence of HAIs, the issue of HAIs is a significant concern, thereby spending hefty sums for controlling and reducing such infections by governments [23].

Although a number of studies have been conducted on different parts of WHO regions to determine the prevalence rate of HAIs, no systematic review has been conducted globally. This study aimed to carry out a systematic review and a meta-analysis to analyze the prevalence of HAIs globally. The research findings will contribute to the development of effective control programs by managers and policymakers of the health sector to reduce the financial costs of HAIs and save financial resources.

## Methods

### Databases and search terms

We conducted a comprehensive search of electronic databases including EMBASE, Scopus, PubMed and Web of Science between 2000 and June 2021. Search terms included ("infection cross"[Title] OR "cross infections"[Title] OR "healthcare associated infections"[Title] OR "healthcare associated infection"[Title] OR "health care associated infection"[Title] OR "health care associated infections"[Title] OR "hospital infection"[Title] OR "infections hospital"[Title] OR "nosocomial infection"[Title] OR "nosocomial infections"[Title] OR "hospital infections"[Title]). We found 7031 articles through searching the databases. After entering the records into EndNote software and removing the duplicates, 5430 studies were screened on the basis of their titles/ abstracts. We reviewed the reference list of all included articles to ensure the comprehensiveness of the search.

### Inclusion and exclusion criteria

On the basis of the research keywords, we included studies reporting quantitative data on HAIs prevalence and their determining factors among the general population. Different observational studies, including cross-sectional, prospective, case-study, and cohort, were included. We considered articles with available full texts published in English between 2000 and June 2021 for further consideration in this review. The reason we included articles from 2000 was to estimate the trend of the current century. We excluded interventional studies, reviews, reports, letters to the editor, books, case-control, and commentaries. We also excluded the review studies using invalid methods or containing insufficient data focused on diagnostic approaches, treatment methods, and medication.

## Study selection

Searching electronic databases resulted in 7031 articles. After removing the duplicates, two researchers reviewed the remaining 5630 records independently, based on the titles and abstracts. In the next step, we systematically evaluated the full texts of the 1909 remaining studies to determine whether they met the eligibility criteria defined in the study. Finally, we selected 400 records with 29159630 participants to evaluate in this meta-analysis (Fig 1).

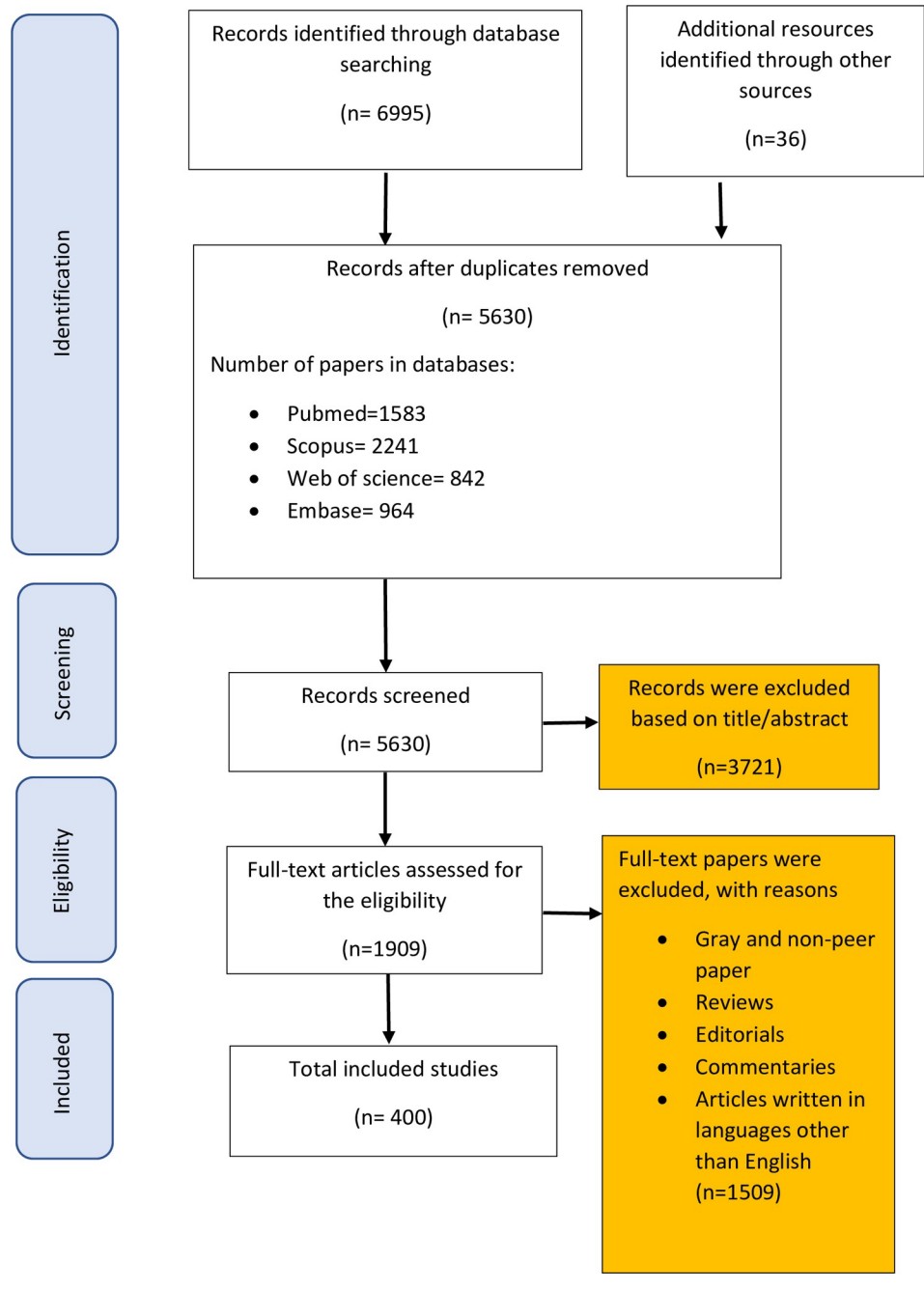

**Fig 1. Flow diagram of our review process (PRISMA).**

## Quality assessment

We evaluated the methodological quality of the articles, using the Newcastle-Ottawa Scale (NOS) based on the procedures suggested in the Cochrane Handbook of Systematic Reviews. The NOS comprises a star system in which a study is evaluated in three areas, including four items regarding the selection of study groups, two items regarding the comparability of groups, and three items in terms of exposure or outcome ascertainment. If any of the items in the NOS were not reported in the article, a zero score was assigned; and for each of the areas addressed in the study, one was given. We categorized studies based on their methodological quality in different groups, from poor (score between 0 and 3) to high quality (score between 7 and 9). Two independent reviewers performed the quality assessment process; in case of any disagreement, a third investigator resolved the issues [24].

## Data extraction

One of the reviewers used a data extraction form to enter data of the included studies. The form included items such as author/ authors' name, the title of the study, publication year, study setting, sample size, characteristics of the study population including their age and gender, the total prevalence of hospital-acquired infection, the prevalence of hospital-acquired infection based on the infection type and related organisms (S1 File).

## Statistical analysis

We used a random-effects model to estimate the pooled prevalence of HAIs, measuring the effect size with a 95% confidence interval (CI) and illustrating the graphical results with Forest plots. The $I^2$ test quantified the statistical heterogeneity, and the Egger test was applied to assess publication bias. We used subgroup analyses due to the variability of estimates based on different study settings, type of infection, and socio-demographic characteristics of study populations. We carried out all analyses, using the Comprehensive Meta-Analysis and R software. All figures with $p<0.05$ were considered statistically significant.

## Patient and public involvement

We considered articles with available full texts published in English between 2000 and June 2021 for further consideration in this review.

# Results

## Overview

According to the inclusion and exclusion criteria and PRISMA checklist (Preferred reporting items for systematic reviews and meta-analyses) [25], we selected 400 articles for the final review stage (see Fig 1). The total number of patients participating in these studies was 29,159,630, of which 5,441,722 had various HAIs. On the basis of the data analysis, we estimated the rate of the global HAIs to be 0.14 (95% CI, 0.12–0.15) (Table 1).

**Table 1. The pooled analysis of global prevalence of nosocomial infection.**

| Models | Number Studies | Effect size and 95% interval | | | Test of null (2-Tail) | |
|---|---|---|---|---|---|---|
| | | Point estimate | Lower limit | Upper limit | Z-value | P-value |
| Random effects | 400 | 0.141 | 0.127 | 0.156 | -32.219 | 0.000 |

As Fig 2 shows, the prevalence of nosocomial infections is increasing, as with a one-year increase, 0.06 would be added to nosocomial infections (Fig 2). Moreover, to clarify the findings, we divided it into five range. The findings show that the highest prevalence of nosocomial infections was 0.20 (95% CI, 0.11–0.32) between 2011–2015, but it decreased to 0.17 (95% CI, 0.08–0.23) between 2016–2011 (Table 6).

Since we included 400 studies in this study and there were different sample sizes, we performed a pooled analysis based on the sample size. The results revealed no significant relationship between sample size and HAIs, and with changing the sample size, we observed no significant difference in the rate of HAIs (Fig 2).

## Meta-analysis based on WHO regions and countries

**Total.** As illustrated in Table 2, the highest rate of HAIs was in the AFR, and based on 94 studies analyzed, this rate was equal to 0.27 (95% CI, 0.22–0.34). The lowest infection rates were in AMR and WPR which were 0.09 (95% CI, 0.07–0.11) and 0.09 (95% CI, 0.06–0.13), respectively. Fig 3 demonstrates the distribution map of HAIs. The map shows that the rate of HAIs in central Africa is higher than anywhere else in the world (Fig 3).

## Meta-analysis based on micro-organism and infection types

Based on the analysis of microorganisms and various HAIs, the findings showed that among all major microorganisms responsible for the HAIs, patients were infected by E. coli more than other microorganisms, 0.18 (95% CI, 0.16–0.20). However, according to WHO regions, *Coagulase-negative staphylococci* was the most common microorganisms in WPRO and EURO with 0.21 (95% CI, 0.11–0.36) and 0.14 (95% CI, 0.10–0.20). Also, in South-East Asian Region Office (SEARO) and EMRO, the highest rate of infection was related to E. coli with

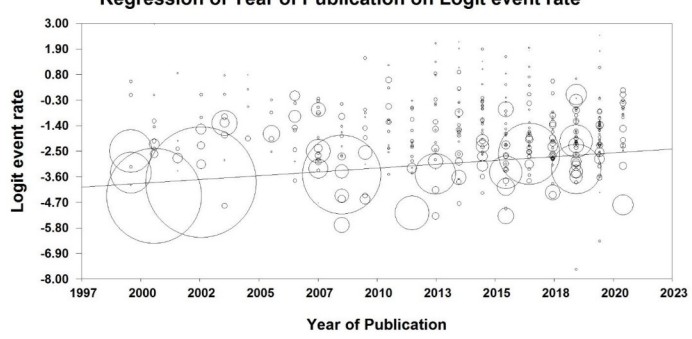

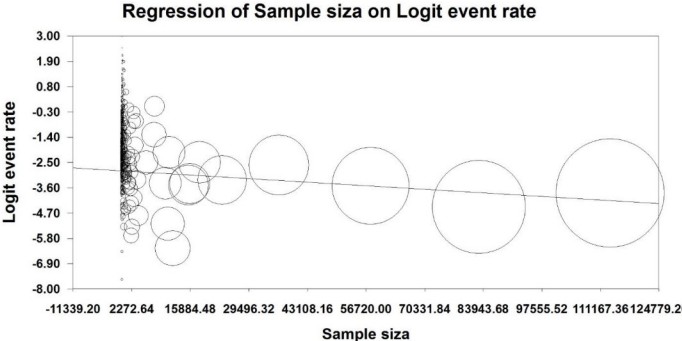

**Fig 2. Meta-regression based on year of publication and sample size.**

**Table 2. Meta-analysis based on WHO regions.**

| WHO Regions | Number Studies | Effect size and 95% interval | | | Test of null (2-Tail) | |
|---|---|---|---|---|---|---|
| | | Point estimate | Lower limit | Upper limit | Z-value | P-value |
| AFRO | 94 | 0.270 | 0.222 | 0.324 | -7.574 | 0.000 |
| AMRO | 18 | 0.096 | 0.079 | 0.117 | -20.059 | 0.000 |
| EMRO | 103 | 0.125 | 0.098 | 0.159 | -13.945 | 0.000 |
| EURO | 114 | 0.114 | 0.096 | 0.134 | -21.394 | 0.000 |
| SEARO | 24 | 0.129 | 0.086 | 0.188 | -8.270 | 0.000 |
| WPRO | 47 | 0.097 | 0.069 | 0.136 | -11.437 | 0.000 |

0.19 (95% CI, 0.13–0.26) and 0.16 (95% CI, 0.13–0.20). Finally, *Pseudomonas aeruginosa* and *Staphylococcus spp.* microorganisms were the most common infectious agents in AMRO and AFRO, respectively (Table 3).

The results of analyzes based on the type of infections showed that the highest type of infection among all HAIs was wound infection, with a rate of 0.34 (95% CI, 0.24–0.47). Regarding the WHO regions, the analyses showed that each region was more involved with a particular infection. For example, in the WPRO and SEARO, respiratory tract infections and surgical site infections were the most common infections. However, wound infection was more prevalent in the EMRO and AFRO than in other infections (Table 3).

## Meta-analysis based on hospital ward

The findings showed that the highest prevalence of HAIs in hospital wards was related to the transplant wards with the prevalence rate of 0.77 (95% CI, 0.38–0.90), followed by Neonatal and ICU wards, with a prevalence rate of 0.69 (95% CI, 0.47–0.85) and 0.68 (95% CI, 0.61–0.73), respectively (Table 4).

## Meta-analysis based on gender

Overall, the prevalence of HAIs is higher in men than women. However, the prevalence of this type of infection is higher in women in AMR and EMR. In AFR, EUR and SEAR, men showed higher prevalence rate, while the rates were the same for both genders in the WPR (Table 5).

## Meta-regression on other sub-groups

**Age.** The results of the analysis showed that the prevalence of HAIs decreases with increasing age. For every one year increase in age, the prevalence decreases by 0.04 (Fig 4). We categorized the participants by age to make our results clear, and we noticed that the highest prevalence of nosocomial infections was in the age range of 0–5 years 0.21 (95% CI, 0.5–0.22) (Table 6).

## Length of stay

According to our findings, No significant relationship was found between length of stay and the prevalence of HAIs (Fig 4). days, We divided the length of stay in the hospital into more than 15 days and less than 15, based on the division of other studies [26]. The results showed that the prevalence rate was estimated to be 0.15 (95% CI, 0.11–0.34) in people who were in the hospital for more than 15 days and 0.12 (95% CI, 0.6–0.28) for those who were in the hospital for 15 days or less than 15 days (Table 6).

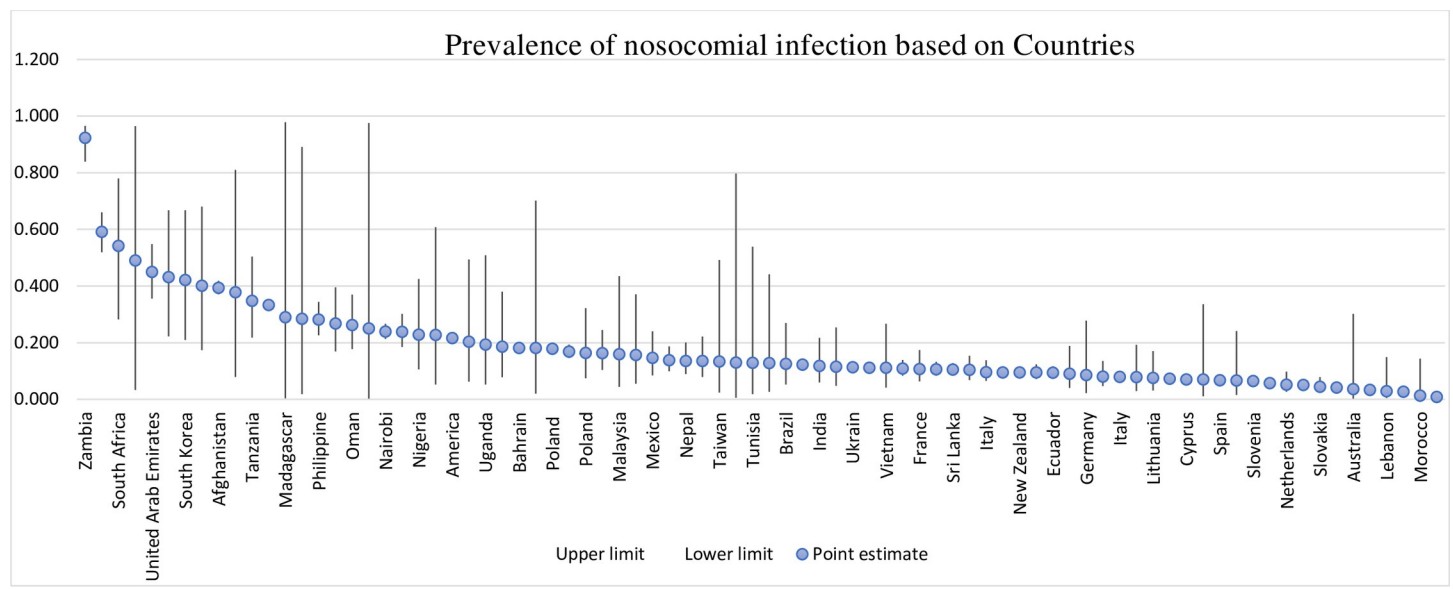

**Fig 3. Distribution of the global prevalence of nosocomial infection in patients based on countries, 2000–2021.** Map created with PhotoshopCC, using political borders.

## Countries based on income

According to the findings of the analysis, countries with lower incomes had higher prevalence of infection. For example, in low-income countries, the prevalence was 0.32 (95% CI, 0.15–

**Table 3. Meta-analysis based on micro-organism and infection types.**

| Sub-groups | | WHO Regions | | | | | | Total |
|---|---|---|---|---|---|---|---|---|
| | | WPRO | SEARO | AMRO | EURO | EMRO | AFRO | |
| **Micro-organism** | Acinetobacter spp | 0.14 | 0.10 | 0.08 | 0.11 | 0.15 | 0.10 | 0.13 |
| | Candida spp | 0.07 | 0.17 | 0.09 | 0.06 | 0.07 | 0.06 | 0.07 |
| | Coagulase-negative staphylococci | 0.21 | 0.04 | 0.15 | 0.14 | 0.15 | 0.11 | 0.14 |
| | E.Coli | 0.10 | 0.19 | 0.12 | 0.05 | 0.16 | 0.24 | 0.18 |
| | Enterobacter spp | 0.08 | 0.15 | 0.11 | 0.07 | 0.09 | 0.07 | 0.08 |
| | Enterococcus spp | 0.05 | 0.12 | 0.08 | 0.06 | 0.07 | 0.04 | 0.07 |
| | Klebsiella spp | 0.13 | 0.14 | 0.12 | 0.12 | 0.12 | 0.14 | 0.13 |
| | Pseudomonas aeruginosa | 0.07 | 0.18 | 0.17 | 0.10 | 0.10 | 0.09 | 0.11 |
| | Staphylococcus spp | 0.14 | 0.17 | 0.63 | 0.13 | 0.13 | 0.26 | 0.14 |
| | Streptococcus spp | 0.03 | 0.01 | 0.06 | 0.05 | 0.06 | 0.04 | 0.05 |
| | Other | 0.04 | 0.17 | 0.03 | 0.08 | 0.12 | 0.04 | 0.05 |
| **Infections** | Bacteraemia | 0.17 | 0.26 | 0.05 | 0.11 | 0.12 | 0.97 | 0.17 |
| | Bloodstream infection | 0.18 | 0.51 | 0.26 | 0.19 | 0.32 | 0.61 | 0.25 |
| | Gastrointestinal infection | 0.08 | 0.91 | 0.46 | 0.07 | 0.08 | 0.93 | 0.12 |
| | Pneumonia | 0.21 | 0.28 | 0.33 | 0.24 | 0.26 | 0.26 | 0.25 |
| | Respiratory tract infection | 0.52 | 0.90 | 0.12 | 0.17 | 0.22 | 0.38 | 0.22 |
| | Surgical site infection | 0.06 | 0.92 | 0.22 | 0.15 | 0.24 | 0.89 | 0.26 |
| | Urinary tract infection | 0.10 | 0.79 | 0.23 | 0.22 | 0.25 | 0.88 | 0.25 |
| | Wound infection | 0.06 | 0.68 | 0.03 | 0.06 | 0.39 | 0.96 | 0.34 |
| | Other | 0.45 | 0.89 | 0.37 | 0.14 | 0.10 | 0.59 | 0.21 |

**Table 4. Meta-analysis based on hospital ward.**

| Hospital Wards | Number Studies | Effect size and 95% interval | | | Test of null (2-Tail) | |
|---|---|---|---|---|---|---|
| | | Point estimate | Lower limit | Upper limit | Z-value | P-value |
| Burns | 19 | 0.25 | 0.15 | 0.38 | -3.60 | 0.00 |
| Cardiology | 2 | 0.07 | 0.06 | 0.08 | -35.16 | 0.00 |
| CCU | 3 | 0.10 | 0.01 | 0.04 | -12.66 | 0.00 |
| Emergency | 9 | 0.22 | 0.09 | 0.46 | -2.28 | 0.02 |
| Hematology | 6 | 0.05 | 0.03 | 0.10 | -7.96 | 0.00 |
| ICU | 140 | 0.68 | 0.61 | 0.73 | 5.37 | 0.00 |
| Internal medicine | 23 | 0.22 | 0.13 | 0.35 | -3.87 | 0.00 |
| Labour & postpartum | 8 | 0.54 | 0.48 | 0.97 | 1.88 | 0.06 |
| Medical wards | 25 | 0.33 | 0.28 | 0.40 | -5.10 | 0.00 |
| Medical wards | 14 | 0.28 | 0.21 | 0.36 | -5.06 | 0.00 |
| Neonatal | 10 | 0.69 | 0.47 | 0.85 | 1.68 | 0.09 |
| Nephrology | 7 | 0.05 | 0.03 | 0.07 | -12.17 | 0.00 |
| NICU | 43 | 0.44 | 0.35 | 0.52 | -1.44 | 0.15 |
| Obstetrics and gynecology | 34 | 0.07 | 0.04 | 0.12 | -9.58 | 0.00 |
| Oncology | 6 | 0.23 | 0.07 | 0.98 | 0.33 | 0.74 |
| Orthopedic | 8 | 0.09 | 0.07 | 0.11 | -15.72 | 0.00 |
| Pediatric | 39 | 0.21 | 0.14 | 0.29 | -5.65 | 0.00 |
| PICU | 21 | 0.28 | 0.19 | 0.39 | -3.83 | 0.00 |
| Rehabilitation | 10 | 0.07 | 0.04 | 0.10 | -11.63 | 0.00 |
| Surgery | 101 | 0.43 | 0.37 | 0.50 | -1.87 | 0.06 |
| Transplant | 5 | 0.77 | 0.38 | 0.90 | 1.72 | 0.09 |
| Trauma | 4 | 0.34 | 0.04 | 0.87 | -0.51 | 0.61 |
| Other wards | 58 | 0.16 | 0.11 | 0.21 | -9.19 | 0.00 |

0.49) and the prevalence of high-income countries was estimated 0.06 (95% CI, 0.03–0.12) (Table 6)

## Quality of study

The findings revealed that the prevalence rate in lower quality studies was 0.16 (95% CI, 0.06–0.19) whereas it was 0.14 (95% CI, 0.10–0.16) in high quality studies (Table 6).

## Publication bias

Based on Fig 5, the analysis showed that this study has a Publication bias. This claim is true since the result of the Egger test was greater than 0.01. (P-value 2-tailed = 0.091).

## Discussion

HAIs are one of the most severe public health issues with high morbidity, mortality, and costs [27]. This study aimed to conduct a systematic review and meta-analysis to determine the prevalence rate of HAIs globally. This is the first comprehensive SLR investigating HAIs from all key aspects. In this systematic review, we screened 7031 journal articles and selected 400 articles that contained quantitative information about the global prevalence of HAIs for evaluation in the meta-analysis.

On the basis of the findings of this study, the rate of universal HAIs is estimated to be 0.14 with an annual increasing rate of 0.06 worldwide. According to our findings, the highest rate

**Table 5. Meta-analysis based on gender.**

| Gender | WHO Regions | Effect size and 95% interval | | | Test of null (2-Tail) | |
|---|---|---|---|---|---|---|
| | | Point estimate | Lower limit | Upper limit | Z-value | P-value |
| Female | AFRO | 0.243 | 0.192 | 0.304 | -7.321 | 0.000 |
| | AMRO | 0.103 | 0.021 | 0.374 | -2.571 | 0.010 |
| | EMRO | 0.252 | 0.176 | 0.348 | -4.652 | 0.000 |
| | EURO | 0.088 | 0.069 | 0.113 | -16.846 | 0.000 |
| | SEARO | 0.145 | 0.103 | 0.199 | -9.028 | 0.000 |
| | WPRO | 0.104 | 0.051 | 0.202 | -5.448 | 0.000 |
| | **Overall** | 0.030 | 0.030 | 0.030 | -1027.257 | 0.000 |
| Male | AFRO | 0.260 | 0.198 | 0.332 | -5.895 | 0.000 |
| | AMRO | 0.067 | 0.024 | 0.177 | -4.722 | 0.000 |
| | EMRO | 0.242 | 0.186 | 0.308 | -6.701 | 0.000 |
| | EURO | 0.114 | 0.091 | 0.140 | -16.751 | 0.000 |
| | SEARO | 0.180 | 0.123 | 0.256 | -6.606 | 0.000 |
| | WPRO | 0.104 | 0.049 | 0.207 | -5.201 | 0.000 |

of HAIs was in the AFR, while the lowest rates were in AMRO and WPRO, at 0.09. Besides, the central Africa had higher rate than other parts of the world. This may be due to the lack of health facilities and resources in this area. The continent is also facing natural crises such as water shortages and droughts, which in turn are increasing nosocomial infections. On the other hand, economic conditions in this region are one of the most important causes of these infections. Another study revealed that the HAIs rate is 7.5% in high-income countries, while it varies between 5.7 and 19.2 in low-income countries percent [28]. According to the WHO data, the HAIs rate is 25% in developing countries and 5–15% in developed countries [29, 30]. Another study estimated the HAIs rate at 16 percent in the Eastern Mediterranean Region [8]. In Roberts et al. study, 159 patients (12.7%) developed HAIs from among 1,253 patients in the United States [31]. The findings of our study suggest that patients are at a higher risk of nosocomial infections due to lack of facilities and poor conditions of hospitals and medical centers in low-income and underdeveloped countries than developed countries.

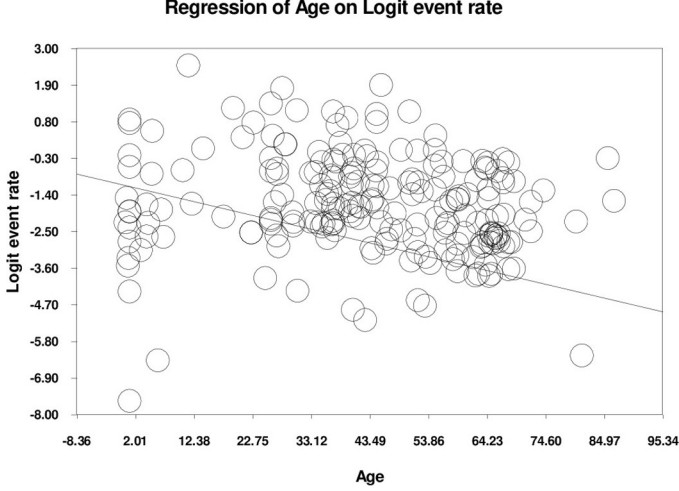

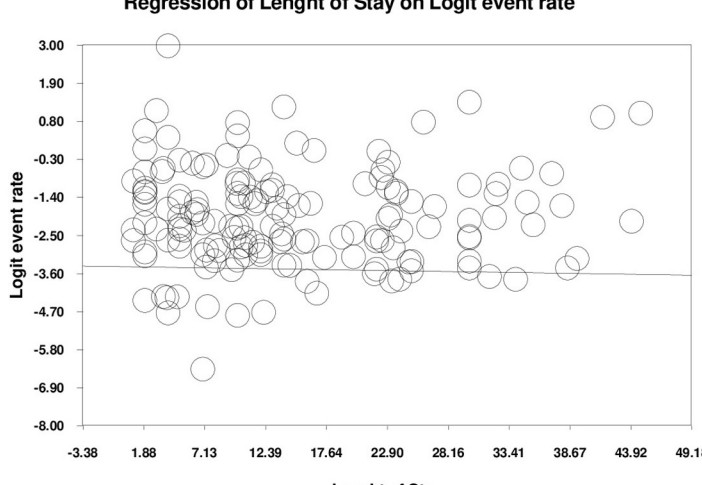

**Fig 4. Meta-regression based on age and length of saty.**

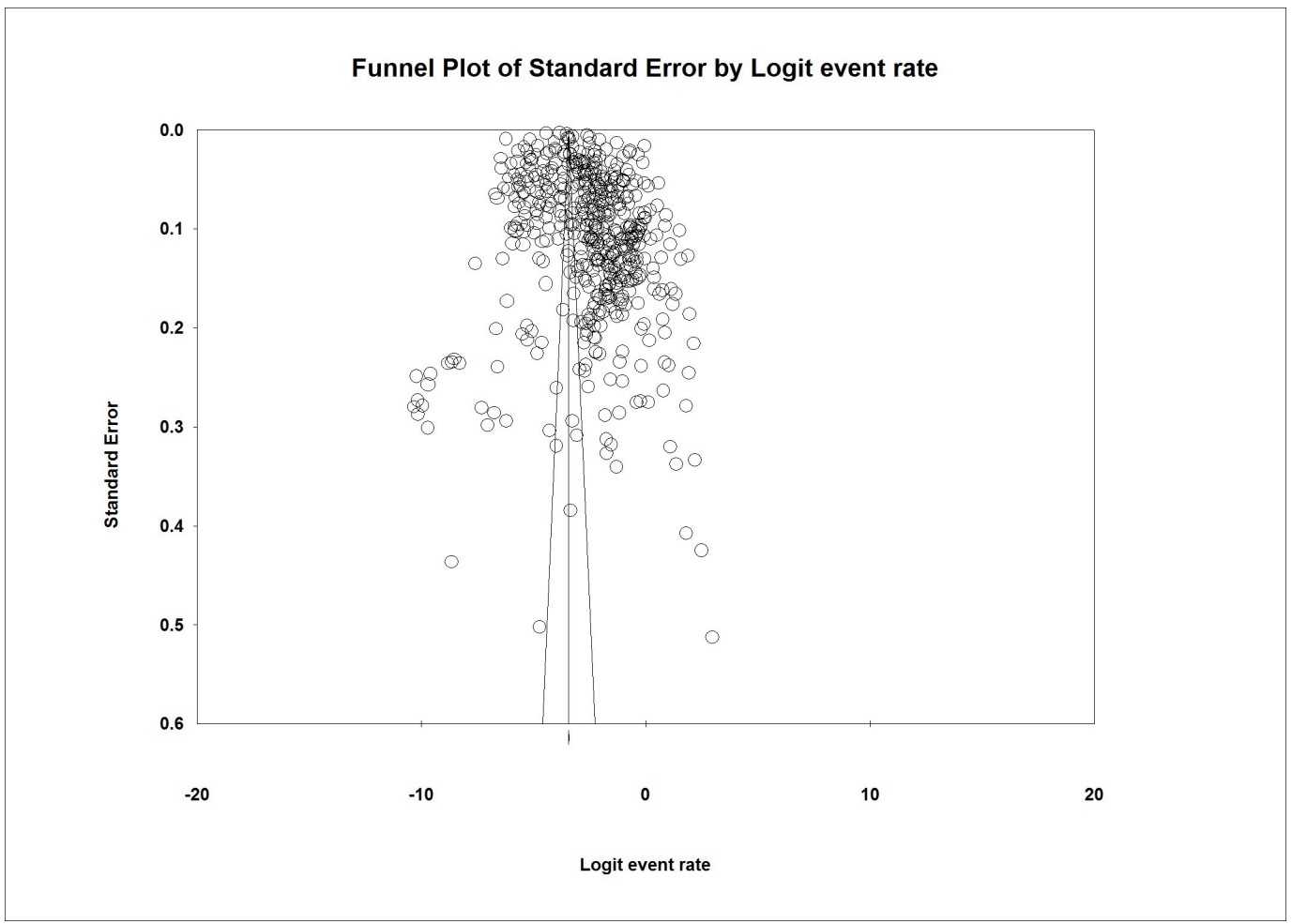

**Fig 5. Funnel plot of publication bias.**

According to the analysis of microorganisms, the E. coli infected patients with HAIs more than other microorganisms (0.18). Based on the WHO regions, *Coagulase-negative staphylococci* were the most common microorganisms in WPRO and EURO with the incidence of 0.21 and 0.14, respectively. In SEARO and EMRO, the highest infectivity was E. coli, with 0.19 and 0.16. Moreover, *Pseudomonas aeruginosa* and *Staphylococcus spp.* microorganisms were the most common infectious agents in AMRO and AFRO, respectively. One of the studies on this issue revealed that *Staphylococcus aureus*, *Pseudomonas aeroginosa*, and *Klebsiella species* are the most common pathogens in Africa and South America [32]. Another review in Africa reported *Klebsiella*, *Staphylococcus aureus*, *Pseudomonas aeroginosa*, and *E.coli* the most common microorganisms in HAIs. These three microorganisms, in addition to being easier to transport than others, have significant resistance to antibiotics. On the other hand, they are more resistant to sterilization and disinfection methods than the others. Due to these characteristics, these microorganisms have a higher prevalence rate than others. Because these bacteria are more resistant to antibiotics than others [33].

Analysis based on the type of infections revealed that the highest type of infection in all HAIs was wound infection, with a rate of 0.34. In terms of the WHO regions, each region represents a specific type of infection; in the WPRO and SEARO. Respiratory tract infections and

**Table 6. The results of subgroups-analysis.**

| Variables | Number of studies | Prevalence (95% CI) | I² | P |
|---|---|---|---|---|
| *Age* | | | | |
| 0–5 | 20 | 0.21 (0.5–0.22) | 97.6% | 0.001 |
| 5–25 | 13 | 0.14 (0.5–0.17) | 98.2% | 0.001 |
| 25–50 | 88 | 0.13 (0.14–0.21) | 98.7% | 0.001 |
| >50 | 67 | 0.17 (0.08–0.19) | 97.4% | 0.001 |
| *Length of stay (Day)* | | | | |
| ≤15 | 85 | 0.12 (0.6–0.28) | 96.6% | 0.001 |
| >15 | 91 | 0.15 (0.11–0.34) | 93.2% | 0.001 |
| *Year of Publication* | | | | |
| 2000–2005 | 78 | 0.11 (0.6–0.19) | 97.2% | 0.001 |
| 2006–2010 | 93 | 0.14 (0.3–0.25) | 96.4% | 0.001 |
| 2011–2015 | 105 | 0.20 (0.11–0.32) | 98.2% | 0.001 |
| 2016–2021 | 124 | 0.16 (0.08–0.23) | 95.7% | 0.001 |
| *Countries based on income* | | | | |
| Low income | 27 | 0.32 (0.15–0.49) | 95.1% | 0.001 |
| Middle income | 26 | 0.16 (0.11–0.26) | 98.3% | 0.001 |
| High income | 29 | 0.06 (0.03–0.12) | 96.5% | 0.001 |
| *Quality of Study* | | | | |
| Low | 26 | 0.16 (0.06–0.19) | 96.4% | 0.001 |
| Medium | 177 | 0.12 (0.05–0.016) | 92.5% | 0.001 |
| High | 197 | 0.14 (0.10–0.16) | 91.1% | 0.001 |

Abbreviations: Confidence Interval (CI).

surgical site infections were the most widespread infections while wound infection was more prevalent in the EMRO and AFRO. A similar study showed that lower respiratory tract infections were the leading cause of HAIs [34]. Other common infections were urinary tract infections, surgical site infections, and bloodstream infections [9].

In terms of the prevalence of HAIs in hospital wards, transplant unit had the highest rate at 0.77, followed by neonatal and ICU wards 0.69 and 0.68, respectively. Nonetheless, a study in Ethiopia found that the infection rate at the surgical site was high % [35]. Another study found the surgical site as the most frequent type of HAI in Low and Middle-Income Countries [9].

Regarding HAIs in terms of gender, the prevalence of HAIs was higher in men than in women. In line with our study, the HAIs burden was shown to be greater in men in another study [36]. In the WHO regions, the rate was higher in women in AMRO and EMRO, whereas, in AFRO, EURO and SEARO, men were reported to have a higher rate. However, in the WPRO, the rates were the same for both sexes. Another study of about 633,000 people in China reported that the prevalence rate was higher in men than in women, supporting our findings [37]. Similarly, another study in the United States on about 530,000 people showed similar results to our findings [10].

With the increasing length of stay in the hospital and despite the fact that we divided the patients into two groups of more than 15 LoS and less than 15 LoS to determine the effect of length of stay on NI, the difference was not too large and we did not find any significant changes in the HAIs rate. However, the AlemkereI'sstudy found that the HAIs risk in patients with a longer stay was 24 times more than in patients with a shorter stay [38]. But in another study, the findings showed that an increase in length of stay could affect the rate of nosocomial infection, but this effect was considered significant after 9 days [39]. In another study of 65,000

people, the findings showed that although longer stays could affect the prevalence of nosocomial infections, the effect was not significant [40]. We think that this variable can affect nosocomial infections, but this effect can manifest itself after a long time stay in the hospital. A short time stay in the hospital does not have much effect and cannot increase the prevalence. According to studies reviewed, this effect becomes more severe after 15 days and increases the prevalence.

The study showed that with the increasing age, the prevalence of nosocomial infections decreases. On the other hand, by age classification, we found that the prevalence of infection is higher in the age group of 0–5 years and in the age group over 50 years. In a study conducted in Argentina on people under the age of five, the prevalence rate of nosocomial infections was reported to be 50%, which was much higher than the average for our study and the global average [41]. In another study conducted in Turkey on people of similar age range, the prevalence rate was about 25 percent [42].

## Conclusion

This systematic and meta-analysis review was conducted to determine the rate of HAIs worldwide. The review identified a number of essential details about the rate of HAIs in various parts of the world. It revealed that the rate of universal HAIs and the number of publications in this regard has risen in recent years. The HAIs rate and the most common micro-organism were different in various regions. However, several important gaps were identified such as lack of data in different regions and territories and different domains like the cause of HAIs. The study findings can help managers and policymakers of the health sector identify the reason for HAIs and apply effective control programs to implement different plans to reduce the HAIs rate and the financial costs of such infections and save resources. We recommend that more studies be carried out to identify strategies and plans for preventing HAIs in all countries, particularly in Low and Middle-Income Countries. Nosocomial infection is one of the most important indicators of hospitals to evaluate the performance of the hospital in terms of patient safety. Our study is done on a global scale so it can be very generalizable and help health decision makers to plan to prevent these types of infections. By reducing nosocomial infections, in addition to improving the patient's safety index, a large amount of the costs incurred by the hospital due to these types of infections will be reduced.

We suggest to decision makers that by focusing on different aspects of nosocomial infections such as age, gender, causes, etc. that we mentioned in this study, comprehensive and practical programs can be used to prevent these infections.

## Limitations

There are some limitations that should be considered when interpreting our study results. First, there might be a language bias in the study as we only included the studies published in English. We focused on peer-reviewed articles; thus, grey literatures and unpublished articles were not included in this review. In addition, in some countries, reliable and published data was not available, so we could not analyze all countries in the world. Finally, studies reviewed did not address many of the variables directly related to nosocomial infections such as type of hospital, number of hospital beds, etc. We also did not include Covid-19 disease in nosocomial infections because they have different definitions, and if we included Covid-19 infections in our study, it would falsely increase the prevalence of nosocomial infections in recent years, it would be a significant bias.

We suggest that researchers work on the gaps in our study. For example, conduct studies in countries where no articles on nosocomial infections have been found. On the other hand,

studies on the cause and transmission of these infections can greatly help the health system to reduce these types of diseases.

## Supporting information

**S1 File.**
(PDF)

**S1 Checklist. PRISMA 2020 checklist.**
(DOCX)

## Acknowledgments

Our research team would like to thank all those who are trying to improve the fields related to health service management, especially the (@health.care.management) team (hcmanagers.ir), who have made great efforts to increase the credibility of this field in the Iranian health system.

## Author Contributions

**Conceptualization:** Ahmad Ghashghaee.

**Data curation:** Fatemeh Pashazadeh Kan, Zahra Hosseinipalangi, Zahra Noorani Mejareh, Saghar Khani, Bahare Abdollahi, Fatemeh Seyghalani Talab, Mohaddeseh Sanaei, Farnaz Zarabi, Yasamin Dolati, Niloofar Ahmadi, Neda Raoofi, Yasamin Sarhadi, Maryam Masoumi, Batool sadat Hosseini, Negin Vali, Negin Gholamali, Saba Asadi, Saba Ahmadi, Behrooz Ahmadi, Zahra Beiramy Chomalu, Elnaz Asadollahi, Mona Rajabi, Dorsa Gharagozloo, Zahra Nejatifar, Rana Soheylirad, Shabnam Jalali, Farnaz Aghajani, Mobina Navidriahy, Sama Deylami, Mahmoud Nasiri, Zahra Golmohammadi, Hamideh Shabani, Fatemeh Torabi, Ahmad Ghashghaee.

**Formal analysis:** Bahare Abdollahi, Negin Gholamali, Behrooz Ahmadi, Zahra Beiramy Chomalu, Ahmad Ghashghaee.

**Funding acquisition:** Ahmad Ghashghaee.

**Investigation:** Fatemeh Seyghalani Talab, Yasamin Dolati, Neda Raoofi, Batool sadat Hosseini, Zahra Beiramy Chomalu, Shabnam Jalali, Farnaz Aghajani, Zahra Golmohammadi, Ahmad Ghashghaee.

**Methodology:** Saghar Khani, Neda Raoofi, Negin Vali, Mona Rajabi, Rana Soheylirad, Shabnam Jalali, Farnaz Aghajani, Sama Deylami, Ahmad Ghashghaee.

**Project administration:** Samira Raoofi, Fatemeh Pashazadeh Kan, Zahra Noorani Mejareh, Ahmad Ghashghaee.

**Resources:** Niloofar Ahmadi, Saba Asadi, Mona Rajabi, Ahmad Ghashghaee.

**Software:** Farnaz Zarabi, Niloofar Ahmadi, Batool sadat Hosseini, Negin Vali, Mobina Navidriahy, Mahsa Zareei, Fatemeh Torabi, Ahmad Ghashghaee.

**Supervision:** Zahra Hosseinipalangi, Zahra Noorani Mejareh, Saghar Khani, Mohaddeseh Sanaei, Farnaz Zarabi, Zahra Nejatifar, Mobina Navidriahy, Sama Deylami, Mahsa Zareei, Ahmad Ghashghaee.

**Validation:** Negin Gholamali, Elnaz Asadollahi, Mahsa Zareei, Fatemeh Torabi, Ahmad Ghashghaee.

**Visualization:** Elnaz Asadollahi, Mahsa Zareei, Aidin Aryankhesal, Ahmad Ghashghaee.

**Writing – original draft:** Samira Raoofi, Sima Rafiei, Hosein Shabaninejad, Mohammad Amerzadeh, Aidin Aryankhesal, Ahmad Ghashghaee.

**Writing – review & editing:** Samira Raoofi, Sima Rafiei, Hosein Shabaninejad, Ali Nemati, Aidin Aryankhesal, Ahmad Ghashghaee.

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
