## [Decision Letter · Decision Letter 0]

7 Mar 2022

PONE-D-22-04059Global, regional, and the national prevalence of nosocomial infection: A systematic review and meta-analysisPLOS ONE

Dear Dr. Ghashghaee,

Thank you for submitting your manuscript to PLOS ONE. After careful consideration, we feel that it has merit but does not fully meet PLOS ONE’s publication criteria as it currently stands. Therefore, we invite you to submit a revised version of the manuscript that addresses the points raised during the review process.

We look forward to receiving your revised manuscript.

Kind regards,

Yong-Hong Kuo

Academic Editor

PLOS ONE

Journal Requirements:

4. We note that Figure 3 in your submission contain [map/satellite] images which may be copyrighted. All PLOS content is published under the Creative Commons Attribution License (CC BY 4.0), which means that the manuscript, images, and Supporting Information files will be freely available online, and any third party is permitted to access, download, copy, distribute, and use these materials in any way, even commercially, with proper attribution. For these reasons, we cannot publish previously copyrighted maps or satellite images created using proprietary data, such as Google software (Google Maps, Street View, and Earth). For more information, see our copyright guidelines: http://journals.plos.org/plosone/s/licenses-and-copyright.

   1. You may seek permission from the original copyright holder of Figure 3 publish the content specifically under the CC BY 4.0 license.  

Maps at the CIA (public domain): https://www.cia.gov/library/publications/the-world- factbook/index.html and https://www.cia.gov/library/publications/cia-maps-publications/index.html

Additional Editor Comments (if provided):

The manuscript has been reviewed by two experts in the area. Both of them find the significance of the study and are positive about the submission. They have provided constructive and very helpful comments to improve the article. Based on their recommendations and comment, I suggest Major Revision.

Reviewers' comments:

Reviewer's Responses to Questions

**Comments to the Author**

1. Is the manuscript technically sound, and do the data support the conclusions?

Reviewer #1: Yes

Reviewer #2: Yes

2. Has the statistical analysis been performed appropriately and rigorously? 

Reviewer #1: Yes

Reviewer #2: Yes

3. Have the authors made all data underlying the findings in their manuscript fully available?

Reviewer #1: Yes

Reviewer #2: Yes

4. Is the manuscript presented in an intelligible fashion and written in standard English?

Reviewer #1: No

Reviewer #2: Yes

5. Review Comments to the Author

Reviewer #1: This is an interesting review article and the authors have collected a unique dataset. The paper is generally well written and organized. However, in my opinion, there are some shortcomings. Some sentences are not well structured and do not bring out the argument clearly. What is the impact of this review to the public health? What recommendations do you have for stakeholders and policymakers?

Reviewer #2: This is a large meta-analysis evaluating the global prevalence of Hospital-acquired infection (HAI). This is an important topic, and the manuscript includes interesting data on causative pathogens for HAI globally and prevalence rates of HAI. The main issue with this manuscript is the number of studies and heterogenous data. With the number of studies, it is difficult to draw broad conclusions. HAI can have different definitions between studies, countries, and institutions. Several co-variates based on different healthcare systems may be unaccounted for. For example, it is surprising that duration of hospitalization was not an associated factor with HAI as seen in previous studies. I would consider limiting the number of studies and potentially decreasing the time the studies were conducted over. With less studies, the manuscript could include a more direct summary and comparison of data from studies.

6. PLOS authors have the option to publish the peer review history of their article (what does this mean?). If published, this will include your full peer review and any attached files.

Reviewer #1: No

Reviewer #2: No

---

## [Author Response · Author response to Decision Letter 0]

22 Jun 2022

Reviews 1 This is an interesting review article and the authors have collected a unique dataset. The paper is generally well written and organized. However, in my opinion, there are some shortcomings. Some sentences are not well structured and do not bring out the argument clearly. What is the impact of this review to the public health? What recommendations do you have for stakeholders and policymakers? 

Answer: Nosocomial infection is one of the most important indicators of hospitals to evaluate the performance of the hospital in terms of patient safety. Our study is done on a global scale so it can be very generalizable and help health decision makers to plan to prevent these types of infections. By reducing nosocomial infections, in addition to improving the patient's safety index, a large amount of the costs incurred by the hospital due to these types of infections will be reduced.

We suggest to decision makers that by focusing on different aspects of nosocomial infections such as age, gender, causes, etc. that we mentioned in this study, comprehensive and practical programs can be used to prevent these infections.

Reviews 2 This is a large meta-analysis evaluating the global prevalence of Hospital-acquired infection (HAI). This is an important topic, and the manuscript includes interesting data on causative pathogens for HAI globally and prevalence rates of HAI. The main issue with this manuscript is the number of studies and heterogenous data. With the number of studies, it is difficult to draw broad conclusions. HAI can have different definitions between studies, countries, and institutions. Several co-variates based on different healthcare systems may be unaccounted for. For example, it is surprising that duration of hospitalization was not an associated factor with HAI as seen in previous studies. I would consider limiting the number of studies and potentially decreasing the time the studies were conducted over. With less studies, the manuscript could include a more direct summary and comparison of data from studies. 

Answer: We explained in the inclusion and exclusion criteria that we only included studies that provided a clear definition of nosocomial infections. On the other hand, the main purpose of our study was to examine the trend of nosocomial infections in the new century, so if the number of articles is reduced, the main purpose of the article will change.

In relation to heterogeneity, we used subgroup analyzes to solve this problem, which were significantly more efficient.

Reviewers Comments Make the title more concise. National and regional prevalence? We changed it to : Global prevalence of nosocomial infection: A systematic review and meta-analysis

Reviewers Comments Results: AFR higher in Central Africa than the world. By how much? We wrote this information in full in the original version of the article, but in the submitted version we had to make corrections due to the word limit.

We added : 0.27 (95% CI, 0.22-0.34)

Reviewers Comments Besides E. coli infected patients…which other microorganisms are you comparing? We added: such as Coagulase-negative staphylococci, Pseudomonas aeruginosa and Staphylococcus spp.

Reviewers Comments Conclusion. Managers? What is their role? Hospital managers and health policy makers

Reviewers Comments Rephrase last sentence in first paragraph. Not clear.

Thus, by preventing the prevalence of HAIs instead of allocating hefty sums to the treatment of such infections, managers of healthcare centers can bear much lower costs to manage HAIs. Therefore, by preventing and reducing nosocomial infections, significant savings will be made in the costs imposed on health centers, the health system and society consequently

Reviewers Comments Which studies?

According to studies, the most prevalent causes of HAIs include urinary tract infections (UTIs), respiratory tract infections (RTIs), circulatory system infections, and surgical site infections. We add a reference for this statement. 

Reviewers Comments Sentence is hanging

Although a number of studies have been conducted on different parts of WHO regions to determine the prevalence rate of HAIs. Although a number of studies have been conducted on different parts of WHO regions to determine the prevalence rate of HAIs, no systematic review has been conducted globally.

Reviewers Comments You need to show up to which date you acquired the 7031 articles. For future references. between 2000 and June 2021

Reviewers Comments Clarify

Increasing rate of HAIs by 0.06% in abstract or 0.6% in results. 0.06 is correct

Reviewers Comments Italicize scientific names We changed them

Reviewers Comments What could result in high HAIs in central Africa? This may be due to the lack of health facilities and resources in this area. The continent is also facing natural crises such as water shortages and droughts, which in turn are increasing nosocomial infections. On the other hand, economic conditions in this region are one of the most important causes of these infections.

Reviewers Comments Why is S. aureus, P. aeroginosa and Klebsiella the most common HAIs? These three microorganisms, in addition to being easier to transport than others, have significant resistance to antibiotics. On the other hand, they are more resistant to sterilization and disinfection methods than others. Due to these characteristics, these microorganisms have a higher prevalence rate than others.

Reviewers Comments Are there any recommendations? We suggest that researchers work on the gaps in our study. For example, conduct studies in countries where no articles on nosocomial infections have been found. On the other hand, studies on the cause and transmission of these infections can greatly help the health system to reduce these types of diseases.

Reviewers Comments Some studies consider COVID 19 as a nosocomial infection. Why did you exclude it? We did not include Covid-19 disease in nosocomial infections because they have different definitions, and if we included Covid-19 infections in our study, it would falsely increase the prevalence of nosocomial infections in recent years, it would be a significant bias.

---

## [Decision Letter · Decision Letter 1]

8 Aug 2022

PONE-D-22-04059R1Global prevalence of nosocomial infection: A systematic review and meta-analysisPLOS ONE

Dear Dr. Ghashghaee,

Thank you for submitting your manuscript to PLOS ONE. After careful consideration, we feel that it has merit but does not fully meet PLOS ONE’s publication criteria as it currently stands. Therefore, we invite you to submit a revised version of the manuscript that addresses the points raised during the review process.

We look forward to receiving your revised manuscript.

Kind regards,

Yong-Hong Kuo

Academic Editor

PLOS ONE

Journal Requirements:

Additional Editor Comments:

There are still minor concerns from the reviewer. Please address them before the final publication.

Reviewers' comments:

Reviewer's Responses to Questions

**Comments to the Author**

1. If the authors have adequately addressed your comments raised in a previous round of review and you feel that this manuscript is now acceptable for publication, you may indicate that here to bypass the “Comments to the Author” section, enter your conflict of interest statement in the “Confidential to Editor” section, and submit your "Accept" recommendation.

Reviewer #1: All comments have been addressed

2. Is the manuscript technically sound, and do the data support the conclusions?

Reviewer #1: Yes

3. Has the statistical analysis been performed appropriately and rigorously? 

Reviewer #1: Yes

4. Have the authors made all data underlying the findings in their manuscript fully available?

Reviewer #1: Yes

5. Is the manuscript presented in an intelligible fashion and written in standard English?

Reviewer #1: Yes

6. Review Comments to the Author

Reviewer #1: This is a well written review paper. The author needs to polish up Discussion section as we do not clearly understand why some microorganisms have a higher prevalence rate as nosocomial infections as compared to others.

7. PLOS authors have the option to publish the peer review history of their article (what does this mean?). If published, this will include your full peer review and any attached files.

Reviewer #1: No

---

## [Author Response · Author response to Decision Letter 1]

9 Aug 2022

Reviews 1 Abstract: 

Objectives- Which needs immediate attention. Specific sounds redundant. “ Specific “ was omitted

Reviews 1 Introduction: 

Remove the irrelevant sentence-It might happen with any kind of infection.

According to studies- Which ones?

Which studies in Europe? Which countries? Any citations?

In the last paragraph, can is subject to speculation. Replace with will All the comments were corrected

Reviews 1 Results: 

Length of stay- Rewrite the first sentence

Countries based on income- Prevalence in high income countries? The first sentence was rewrote. 

The prevalence of high-income countries was added

Reviews 1 Discussion- 

Why is certain organisms rated as more common HAIs than others? Because these bacteria are more resistant to antibiotics than others.

Reviews 1 Conclusion- 

Which gaps? However, several important gaps were identified such as lack of data in different regions and territories and different areas like the cause of HAIs

---

## [Decision Letter · Decision Letter 2]

25 Aug 2022

Global prevalence of nosocomial infection: A systematic review and meta-analysis

PONE-D-22-04059R2

Dear Dr. Ghashghaee,

We’re pleased to inform you that your manuscript has been judged scientifically suitable for publication and will be formally accepted for publication once it meets all outstanding technical requirements.

Kind regards,

Yong-Hong Kuo

Academic Editor

PLOS ONE

Additional Editor Comments (optional):

Based on the Referees' recommendations, I recommend Accept.

Reviewers' comments:

Reviewer's Responses to Questions

**Comments to the Author**

1. If the authors have adequately addressed your comments raised in a previous round of review and you feel that this manuscript is now acceptable for publication, you may indicate that here to bypass the “Comments to the Author” section, enter your conflict of interest statement in the “Confidential to Editor” section, and submit your "Accept" recommendation.

Reviewer #1: All comments have been addressed

2. Is the manuscript technically sound, and do the data support the conclusions?

Reviewer #1: Yes

3. Has the statistical analysis been performed appropriately and rigorously? 

Reviewer #1: Yes

4. Have the authors made all data underlying the findings in their manuscript fully available?

Reviewer #1: Yes

5. Is the manuscript presented in an intelligible fashion and written in standard English?

Reviewer #1: Yes

6. Review Comments to the Author

Reviewer #1: It is an interesting article. This is a comprehensive meta-analysis that looks at the global prevalence of hospital-acquired infections (HAI). This is an essential issue, and the paper contains intriguing statistics on HAI causative microorganisms and HAI prevalence rates throughout the world. Submitted comments have been addressed.

7. PLOS authors have the option to publish the peer review history of their article (what does this mean?). If published, this will include your full peer review and any attached files.

Reviewer #1: No

---

## [Editor Report · Acceptance letter]

18 Jan 2023

PONE-D-22-04059R2 

Global prevalence of nosocomial infection: A systematic review and meta-analysis 

Dear Dr. Ghashghaee:

I'm pleased to inform you that your manuscript has been deemed suitable for publication in PLOS ONE. Congratulations! Your manuscript is now with our production department. 

Kind regards, 

on behalf of

Dr. Yong-Hong Kuo 

Academic Editor

PLOS ONE